# Understanding the pathways to text generation: A longitudinal study on executive functions, oral language, and transcription skills from kindergarten to first grade

**Juan E. Jiménez** [1]*, **Jennifer Balade**[1], **Eduardo García** [1], **Becky Xi Chen**[2]

**1** The University Institute of Neuroscience (IUNE), Universidad de La Laguna, The Canary Islands, La Laguna, Spain, **2** Department of Developmental of Applied Psychology and Human Development, Ontario Institute for Studies in Education, University of Toronto, Toronto, Canadá

* ejimenez@ull.edu.es

## Abstract

This longitudinal study explored the contribution of transcription skills, oral language abilities, and executive functions in kindergarten to written production in grade 1 among Spanish-speaking children (N = 191) through structural equation modeling (SEM). Three dimentions of written production were assessed, including productivity, quality, and syntactic complexity. Accordingly, three SEM models were tested to explore these relationships, and the estimated models for each endogenous variable demonstrated good fit. The results indicate that transcription skills and executive functions were key predictors of productivity, while both transcription and narrative oral competence contributed to writing quality. Syntactic complexity, on the other hand, was primarily influenced by narrative oral competence and executive functions. The results are interpreted within the framework of the not-so-simple view of writing model, particularly considering the characteristics of a shallow orthography. Limitations and educational implications are also discussed.

## Introduction

Proficient writing is crucial for academic and professional success [1]. Given its importance, writing has received significant research attention for supporting children's development [2]. Acquiring writing skills is a fundamental yet challenging aspect of literacy development during the elementary school years, necessitating mastery of several critical component skills such as grasp of syntaxis, lexical acquisition, coherent ideation, rigorous editing and revision practices, and strategic discourse organization [3,4]. A primary approach to studying writing development involves analyzing various textual factors to uncover the dimensions underlying the variability in children's writing [5–7]. This approach acknowledges that writing production is a multidimensional phenomenon and highlights the insufficiency of focusing solely on one aspect, such as text quality [6,8]. In early childhood, writing production is increasingly recognized as a complex process that involves not only the final quality of the written text but also

**Data Availability Statement:** The database has been uploaded as a Supporting Information file.

**Funding:** Grant PID2019-108419RB-100 funded by MCIN/AEI/10.130.39/50110001103.

**Competing interests:** The authors have declared that no competing interests exist.

factors like productivity and syntactic complexity. Kim et al. [8], in their study on the dimensionality of first-grade written composition, identified these three dimensions as critical for evaluating writing development. Productivity, which refers to the amount of text produced (e.g., total words), reflects a child's ability to generate content fluently. Quality captures higher-order features of writing, such as organization, coherence, and clarity of ideas, while syntactic complexity refers to the sophistication of sentence structures and grammatical accuracy. While Kim et al. (2014) provided valuable insights into the dimensionality of writing in English-speaking first-grade students, there remains a gap in understanding how these dimensions (productivity, quality, and syntactic complexity) manifest in other linguistic contexts, such as Spanish—a language with a shallow orthography. Extending this research to Spanish-speaking students allows us to explore whether similar patterns hold in different linguistic environments, despite differences in orthographic depth between languages. Furthermore, our study contributes to the field by examining how transcription skills, oral language abilities, and executive functions interact to influence these dimensions of writing, thus providing a more comprehensive understanding of early writing development.

## Writing development models: Individual skills in early text production

Given the significance of acquiring writing skills, it is imperative to explore current research approaches for understanding writing development. Developmental models of writing, such as the simple view of writing [9] and its extension, Berninger's not-so-simple view of writing (NSVW) [10,11], suggest that written text production is influenced by three closely connected component skills: transcription skills, oral language skills, and executive functions. These components play a crucial role in shaping the process of written text production. These skills interact dynamically with working memory during the writing process, shaping its progression [12]. In the model NSVW, transcription skills refer to the mechanical process of converting ideas into written symbols, such as spelling and handwriting. Oral language proficiency plays a crucial role in the text generation phase, facilitating the conversion of ideas into linguistic forms stored in working memory. Executive functions oversee and regulate the writing process, facilitating attention, planning, and revision. Thus, the early stages of writing require transcription skills, executive functions and oral language skills [13,14].

As mentioned above, transcripton skills consist of handwriting and spelling. Handwriting involves motor movements, while spelling entails symbol retrieval and assembly [15,16]. These skills are vital for quality and productivity in writing development as early as in kindergarten [17], aiding language-to-text transformation [10,18,19]. The development of transcription skills continues throughout schooling, contributing to enhanced writing proficiency [7,20,21]. Although both skills positively correlate with writing performance, their impact varies across educational levels [20]. In primary school, they directly influence writing, while in middle school, their influence is indirect through planning and self-efficacy [22].

Previous research has produced mixed findings about the relationships among handwriting, spelling, and composition. Although comprehensive analyses by Santangelo and Graham [23] revealed notable improvements in writing quality, length, and fluency following handwriting fluency training, the longitudinal connections between handwriting and composition have been somewhat limited [24]. For example, Graham and Santangelo [25] observed a minimal impact of spelling instruction on overall writing performance, in addition to spelling accuracy, which contrasts with the findings of Abbott et al.'s [24] longitudinal investigation where spelling emerged as a robust predictor of composition quality. As noted by Vieira et al. [4], these contrasting findings may arise from methodological disparities across studies,

encompassing participant characteristics, evaluation metrics, study design, linguistic factors, cultural influences, and pedagogical methodologies.

According to the not-so-simple view of writing, both text generation and transcription skills are core processes in writing. Oral language skills, which are essential to the text generation process, serve as significant predictors of written expression [26–31]. In addition to their direct impact on text generation, oral language skills also exert an indirect impact by reducing the processing demands on working memory when fluency in idea translation is achieved [32,33]. Evidence supports the contribution of oral language skills to writing in children across various grades, with syntax instruction shown to enhance writing performance [34,35].

Notably, the findings have been mixed across different grades in opaque orthographies. While some studies have reported this relationship in first-grade children [28], the majority have found it from the 2nd or 3rd grade onward [1,18,26,36]. In a classic study conducted by Juel et al. [28], ideation, as assessed by a storytelling task, contributed significantly to writing in Grades 1 and 2. However, Abbott and Berninger [18] and Berninger and Abbott [26] found that oral language skills significantly contributed to English written composition in grades 2 to 3 and grades 6 to 7 but not in grade 1 or intermediate grades. Yeung et al. [1] observed similar results in Chinese written composition, with Grade 1 oral language skills predicting Grade 2 composition but not Grade 1. In their meta-analysis, Kent and Wanzek [36] examined the correlation between various proposed component skills, including handwriting fluency, spelling, reading, and oral language, and writing outcomes. They evaluated both the quality of students' compositions and the quantity of their written output. The results indicated that oral language exhibited the weakest relationship with writing quality, explaining slightly more than 10% of the variance. Nevertheless, longitudinal evidence regarding the contribution of oral language skills to writing is limited [16]. While some studies suggest that oral language skills assessed in the early grades may predict narrative writing growth over time [35], others find a relationship between oral vocabulary proficiency and writing quality and quantity from the outset [37]. These findings underscore the complex interplay between oral language skills and written composition, highlighting the need for further research in this area.

While much research has focused on transcription and oral language skills, there is a need for further exploration of the role of executive functions in writing development [16]. Although several studies have been conducted in this area, much less is known about the impact of executive functions on writing outcomes compared to that of transcription and oral language skills [13,14,16,38]. Executive functions (EFs), including working memory, inhibitory control, and cognitive flexibility, are essential cognitive processes for goal-directed behavior and problem solving [39]. EF encompasses both higher-level cognitive functions involved in planning, translating, reviewing, and revising to regulate the writing process, as well as foundational skills supporting these processes [40,41]. Foundational EFs are considered cognitive subcomponents of a single supervisory attentional mechanism, typically involving inhibition, working memory updating, and shifting [42]. Inhibition involves selectively attending to specific stimuli while suppressing attention to others (selective attention), maintaining task focus despite distractions (sustained attention), and inhibiting prepotent responses. Working memory updating entails retaining pertinent information, such as representational structures, mentally manipulating it, and taking actions based on it. Finally, shifting, or cognitive flexibility, involves adapting swiftly and flexibly to changing situations, such as tasks or mental sets [43].

These functions are critical for writing and influence various aspects of composition, including transcription and text quality [44–46]. Working memory facilitates the storage and manipulation of information during composition, aiding in planning, language generation, and review [45]. Inhibitory control helps writers suppress irrelevant ideas and select

appropriate words and structures [47], while cognitive flexibility allows for switching between perspectives [16].

However, studies examining the relationship between EFs and writing in primary school children have presented mixed findings [13,16,35,38,48–50]. For instance, WM measured in Grade 2 predicted text quality six months later, but its direct relation to writing was not observed in Grade 1 [13,50]. Inhibitory control is longitudinally related to syntactic complexity in narratives [16], while inhibitory control and cognitive flexibility contribute to spelling and written expression [48]. Moreover, attentional processes, as emphasized in studies by Kent et al. [8] and Hooper et al. [35], are pivotal in early writing, facilitating students' focus on relevant tasks, retention of prior content for future composition, and filtration of irrelevant information. Valcan et al. [38] analyzed the contribution of EF to handwriting composition in Year 2 children using the HTKS, a task that integrates multiple EF components (i.e., working memory, inhibition, and attention shifting). They found that handwriting automaticity and spelling mediated the relationship between children's EF and handwriting composition (i.e., fluency and quality). Kent et al. [49] demonstrated that attention predicted both composition quality and fluency in the first grade, while Hooper et al. [35] identified attention/executive function as the primary predictor of first-grade writing. These findings emphasize the significance of integrating attentional mechanisms into our understanding and prediction of early writing development, justifying its inclusion as an EF latent factor in our study. In contrast, in recent studies with primary and middle graders, both inhibitory control and cognitive flexibility failed to predict text quality [13]. Moreover, findings regarding the most predictive EF vary across studies, indicating a lack of consensus in the field [35,51].

One reason for the failure to establish a clear link between EF and the quality of written production may stem from the initial unitary development of EF, which gradually differentiates into distinct components [16,35,38]. EF functions as a unified construct during the ages of 3 to 8 [52]. It evolves into three interrelated but distinct components as children grow older [42]. This developmental trajectory suggests that analyzing EF as a unified construct may be more advisable, especially when dealing with kindergarten students, as in the present study.

## Examining transcription, oral language, and executive function skills on dimensions of early writing production in a shallow orthography

First, it has been proposed that the relationships among handwriting fluency, spelling proficiency, and narrative writing may be stronger in opaque orthographies such as English [7,9,24,53–55] than in shallow orthographies like Spanish. Opaque orthographies exhibit less predictable correspondences between letters and sounds, potentially increasing the demand for spelling during text production. Conversely, in shallow orthographies, where the correspondences between letters and sounds are more consistent, the demand for spelling during text production is reduced (e.g., in Dutch, [16,55,56]; in Finnish, [57]; in Italian, [3,5,58]; in Icelandic, [59]; in Norwegian, [21]; in Portuguese, [4,22]; in Spanish, [60–63]; and Turkish, [30,32]). As noted by Longobardi et al. [3], studies conducted in shallow orthographies generally do not support the role of the transcription process in writing. Taken together, this body of research suggests that children acquiring languages with more uniform orthographies exhibit decreased dependence on transcription skills, specifically word-spelling abilities, compared to those learning languages with irregular orthographic patterns.

Oral language skills play a critical role in the development of writing [35], which requires children to translate ideas into linguistic forms, organize thoughts coherently, and employ an extensive vocabulary [54,64]. In languages with shallow orthographies, the demands on spelling during text production are reduced due to consistent correspondences between letters and

sounds [5,65]. Studies conducted with Italian children, such as the research by Arfé et al. [5], demonstrate that oral language skills contribute to microstructural productivity, complexity, accuracy, and overall text quality among Italian second and third graders. Receptive grammar skills have a more significant impact on the macrostructural level of written composition than spelling. Similarly, oral narrative competence directly affects the written narrative production of second graders [58]. In a longitudinal study with Icelandic children, Oddsdóttir et al. [59] assessed text generation skills in first grade and narrative and informational text composition in second and fourth grades. They found that children activated their text generation skills earlier in narrative writing than in information text writing. Given that our study followed children from kindergarten to grade 1, we chose to focus on narrative writing.

Research has shown that while the role or influence of oral language skills remains consistent across different languages [30,35], differences emerge in the timing of this influence, with an earlier onset observed in shallow orthographic systems due to children's precocious automation of transcription skills [66]. In opaque orthographies, the scenario is different. For instance, Hooper et al. [35] reported that oral language skills assessed prior to kindergarten entry predicted the developmental trajectory of narrative writing from third to fifth grade, suggesting that the predictive power of oral language skills emerges in later grades. In a longitudinal study that followed New Zealand children from kindergarten to grade 1, McDonald et al. [66] found that the link between oral language skills and writing proficiency emerged later once transcription skills were more consolidated. Similarly, in the studies cited above (e.g., [1,18,26]), the lack of significance of oral language skills in Grade 1 was attributed to the need to first overcome the constraints imposed by transcription skills before oral language skills could exert influence.

Although there is less emphasis on spelling accuracy in languages with shallow orthographies, young writers still face challenges arising from the intricacies of grammar and morphology, as highlighted by previous studies [5,67,68]. Moreover, research suggests that children learning to write in languages with shallow orthographies may rely more heavily on executive functions to manage the cognitive demands of text production, particularly in navigating the complexities of grammar and morphology [4,13,16]. For instance, Drijbooms et al. [16] observed a longitudinal relationship between EFs, specifically inhibition and planning, and syntactic complexity in written narratives among Dutch students in grades 4 and 6. Similarly, Cordeiro et al. [13] reported that among Portuguese Grade 2 students, working memory and planning at the beginning of the school year made a significant contribution to text quality six months later, surpassing the influence of transcription skills. Vieira et al. [4] reported that EFs were associated with text quality in Grade 3, pointing to the central role of EFs in written expression, especially among older children.

Several previous studies conducted in Spanish have examined how transcription and oral language skills contribute to the productivity and quality of written compositions in young children [60–62]. For instance, in a study that involved Spanish students in grades 1 and 2, Jiménez and Hernández-Cabrera [60] engaged children in a conversation about their interests, favorite games, hobbies, among other topics, and after the child described one or two situations, they were asked to compose two sentences about the conversation. The total number of words written correctly within one minute during these tasks served as an index of writing fluency. The results showed that handwriting fluency and spelling directly contributed to writing fluency in sentence composition. Notably, handwriting fluency contributed less variance than spelling to writing fluency in this study. Spelling contributed to both narrative writing and writing fluency. Jiménez and Barrientos [61] studied the impact of transcription skills, assessed through graphonomic measures, on text generation proficiency in primary school students in Spain. They used a longitudinal design with 278 students across three cohorts (cohort 1: 1st-

2nd-4th grade; cohort 2: 2nd-3rd-5th grade; and cohort 3: 3rd-4th-6th grade). Four multi-group structural equation models were used to examine direct pathways from graphonomic measures (e.g., pressure, speed, pauses, and road length) to text generation (e.g., length, fluency, planning, revision, and organization). The results showed that handwriting significantly influenced text production in Cohort 1 (early grades) but not in Cohort 2 (intermediate grades) or Cohort 3 (upper grades). Recently, Rodríguez et al. [62] explored the relationships among transcription skills, oral language abilities, and writing quality and productivity in Spanish-speaking kindergarten children. The results showed that transcription skills strongly predicted both writing quality and productivity, while oral language only predicted writing quality. However, the limitations of the previous studies on Spanish include the sole focus on transcription skills without assessing handwriting [62], reliance on observed variables instead of latent variables [62], and the absence of oral language and executive function assessments (e.g., [60,61]). The present study aimed to address these limitations by incorporating handwriting assessment while considering oral language skills and executive functions in a longitudinal design.

## The present study

This study examined the contribution of transcription skills, oral language abilities, and executive functions in kindergarten to written production in grade 1 among Spanish children. The Spanish language exhibits a relatively shallow orthographic system compared to other languages (such as French and English), where morphology is more extensively represented [64]. Despite respecting some aspects of morphology, such as the use of 'h' in verb forms such as "haber," Spanish orthography is relatively limited in its representation of morphology. On the other hand, Spanish orthography is characterized by its consistent and efficient representation of phonological information, making it relatively easier to learn to read and spell compared to languages with more complex orthographic systems [69].

Previous studies in languages with shallow orthographies have shed light on the unique challenges posed by these writing systems. While there is less emphasis on spelling accuracy in shallow orthographies, young writers still encounter difficulties stemming from the complexities of grammar and morphology [5,67,68]. Similarly, the role of transcription skills, oral language skills, and executive functions has also been studied in the development of writing in shallow orthographies. However, comprehensive investigations encompassing all these facets within a single study, especially targeting kindergarten-aged children, are notably lacking. Therefore, the present study not only extends previous research to a younger age group but also provides new insights by examining how these three sets of skills influence different dimensions of written production (quality, productivity and syntactic complexity) in a shallow orthography. We expected that in a shallow orthography such as Spanish, transcription skills may be less predictive of textual dimensions such as productivity, quality and syntactic complexity than oral language skills and executive functions. In other words, considering the nature of the orthography, oral language skills and executive functions are expected to play more significant roles in shaping written production.

## Method

### Participants

A total of 198 Spanish students participated in the study. Participants were recruited from seven schools in the Canary Islands during their final year of kindergarten. Children with special educational needs, such as sensory, motor, or intellectual impairments, as well as those with established diagnoses such as Down syndrome or autism spectrum disorder, were

excluded from the analysis. The final sample comprised 191 children (96 boys and 95 girls; kindergarten age M = 5.40; SD = 0.28; 1st grade age M = 6.92; SD = 0.28). The sample included students from both state and nonstate schools, ensuring socioeconomic diversity. Approximately 50.2% of the families in the sample provided responses regarding their educational and socioeconomic levels. For mothers, 27.1% held a secondary education (Bachillerato) diploma, 29.2% held a university degree, and 13.5% completed vocational training. For fathers, 30.2% held a secondary education (Bachillerato) diploma, 18.8% held a university degree, and 14.6% completed vocational training. With respect to socioeconomic status, the majority of the mothers (56.3%) reported middle-income levels, while 24% fell into the low-income category. A smal percentage, 6.3%, reported having a high income, and 9.4% of the mothers reported having no income. Similarly, for fathers, the largest group (61.5%) also reported middle income levels. Fathers in the low-income category constituted 20.8%, while those in the high-income category made up 4.2%. Additionally, 1% of fathers reported having no income. All participants were native speakers of Spanish, and due to the cultural and linguistic context of the Canary Islands, the sample consisted exclusively of individuals who spoke Spanish as their first language.

## Materials

Assessments of executive functions, transcription, and oral language were conducted in kindergarten. Subsequently, when the students reached the the first grade, assessments of narrative writing were conducted.

**Transcription skills.** *Phoneme Isolation.* This task was designed to assess the child's ability to phonemically isolate the initial sound of each word heard. This phonemic awareness skill is crucial for kindergarten children learning to read in a shallow orthography (e.g., [70–72]). The child was asked to listen to a word and say the first sound of the word. The task included 18 words. Two points were assigned for each phoneme correctly identified, and one point was given if the phoneme was identified by the corresponding letter name. The Cronbach's alpha for this sample was 96.

*Letter copying.* This measure of handwriting fluency assessed children's proficiency in reproducing letters. The copy letter task, administered within one minute, has been used to assess handwriting automaticity in children [7,54,73]. The child was asked to copy the vowel letters that appeared on a screen one by one within one minute. The score was the number of correctly copied letters within that time frame. Correct letters were those that did not contain errors, such as deletion, misalignment, addition, or inversion. The average intraclass correlation coefficient (ICC), which is an indicator of interrater reliability for handwriting measures, was .960, and the 95% confidence intervals ranged from .948 to .969 (F (214, 214) = 49.2, $p <$ .001).

*Name writing task.* This task has been extensively studied in previous research (e.g., [74–78]). The task consisted of two parts. In the first part, the participant was asked to write their own name, and in the second part, they were asked to write the names of their friends, classmates, family members, among others. Once the second part was completed, the child was asked to state what they wrote on each line. The number of correct words was recorded. A word was considered correct only if it contained all the letters. The use of homophones was not penalized (e.g., Hugo vs ugo), nor was the use of uppercase or lowercase letters.

*Picture Word Writing.* This task is well-suited for assessing spelling in young children due to its demonstrated appropriateness in early-age assessments [79–81]. The children were presented with a total of 12 pictures and asked to write the names of the depicted objects. Legible letters were counted, with no penalty for the use of uppercase or lowercase letters.

Homophones were accepted (e.g., 'nube' vs 'nuve'). The total score represented the number of correctly spelled letters. The Cronbach's alpha for this task for this sample was 0.91.

*Phoneme Segmentation.* This task was adapted from a well-known task developed by Liberman et al. [82]. However, in our adaptation, children were presented with pseudowords instead of real words to minimize the potential influence of orthographic knowledge. The examiner pronounced a pseudoword, and the child was required to segment it by identifying its phonemes. Phonemes that were correctly segmented were recorded. The maximum possible score of this task was 85 points. The Cronbach's alpha for this task for this sample was 0.97.

**Oral language skills.** *Oral Narrative Task.* A story production task was employed for evaluating oral narrative competence. This task has been used in young children in previous studies (e.g., [83,84]). The children were asked to narrate a story based on a picture provided. The image depicted an ice cream cart with a vendor and two additional characters: a child who dropped his ice cream on the ground and an adult who purchased the ice cream. The evaluation covered various aspects: a) structure: children were scored 0, 1, or 2 points for each story structure component outlined in the rubric (title, opening, characters, main event, problem, emotion, resolution and closure); b) the count of unique words; c) the total number of T-units; and d) the total number of words.

For narrative task (structure), the average ICC was .710, with a 95% confidence interval from .644 to .766 F (256, 256) = 5.90, p < .001. For single-word writing, the average ICC was 0.96, with a 95% confidence interval ranging from 0.94 to 0.97. Statistical significance was observed, with F (114, 115) = 23.0 and p < 0.001. For T-units and grammatical structure, the average ICC was 0.85, with a 95% confidence interval ranging from 0.79 to 0.89. A statistically significant difference was observed, with F (114, 115) = 6.6 and p < 0.001.

**Executive functions.** *Perception of Differences Test (Caras-R)* [85]. This task assessed attention processes with a total of 60 sets of items. Each item comprised three drawn faces: two of them were identical, while the third differed only in a minor detail (for example, open eyes vs. closed eyes). Participants were required to select the stimulus that was different from the other two options. The score was determined by the total number of correct responses within a 3-minute period. According to the test manual, the test-retest reliability was .60, and the split-half reliability was 0.97.

*Digit Span Backward.* This task was adapted from the digit span task of the WISC-V [86]. The task assessed students' working memory. It was divided into two blocks: forward and backward. Only the backward block was used in the present study. In the backward block, students were asked to listen to a sequence of numbers and name the numbers in reverse order. For example, if they heard 8–7, they should respond with 7–8. The number of digits in the sequence increased every four sequences. The total number of sequences in this block was 18. The score represented the total number of correct sequences in the backward block. The Cronbach's alpha for this sample was 78.

*Oral Cloze Task.* Designed to assess verbal working memory, this task was an adaptation of a task used by Siegel and Ryan [87]. The experimenter aurally presented a set of incomplete sentences with the final word missing. The student was asked to complete the sentences by supplying the final word and repeat the missing words in the same order that the sentences had been presented. Prior to starting the task, the student was given a practice set with two sentences and two attempts for the practice item. The task consisted of six sets with three attempts for each set. The first set consisted of two sentences, and set six had seven sentences. The task was discontinued when the student failed all attempts at the same level. The score was based on the total number of correctly remembered attempts, with a maximum score of 18 points. The Cronbach's alpha for this sample was 73.

*Inhibitory Control*. This task was adapted from the Stroop day–night task [88]. It assessed the ability to inhibit automatic responses based on semantic content and to process visual information accurately. In this version, which consisted of two main blocks, we used items depicting suns and moons. The first block contained the congruent condition in which students were asked to say "sun" when they saw the sun and "moon" when they saw the moon. In the second block, the incongruent condition, the students were asked to say "sun" when they saw the moon and vice versa. In each block, a total of 50 items were presented, arranged in a matrix of 5 rows by 10 columns. Ten practice items were used for each of the two conditionsto ensure task comprehension. The children were instructed to respond as quickly as possible in both conditions within 45 seconds. The score was calculated by dividing the total number of correct answers by the time in the incongruent block. The test-retest reliability for the incongruent condition was 91 [89].

*Cognitive Flexibility*. Adapted from the Dimension Change Card Sort Task [90], this task assessed cognitive flexibility, which is the ability to adjust behavior in response to changes in rules and instructions. Students were presented with two options at the bottom of the screen: a blue cat and a red car. The target items are shown at the top. This task consisted of three main blocks: 1) Prechange. Students were asked to classify the target items according to either color or shape. Classification by one dimension or the other was counterbalanced; half of the students started with the color condition, and the other half started with the shape condition. In the color condition, when the target item was a blue car, the participants clicked on the blue cat, and if a red cat appeared, they clicked on the red car. In the shape condition, when the target item was a red cat, the participants clicked on the blue cat in the bottom section, and when the target item was a blue car, they clicked on the red car in the bottom section. 2) Postchange. In this block, the students were asked to switch to the other condition. That is, those who classified items by color in the previous block classified by shape in this block and vice versa. 3) Border. In this block, the target items may be presented either with or without a black border. Children were instructed to play the color game when a card contained a border and the shape game when the cards did not have a border (or vice versa if the prechange condition had a shape dimension). This task consisted of a total of six items for the prechange block, another six items for the postchange block, and 14 items for the border block. The score used was a fluency index, which was calculated by dividing the total number of correct answers by the time in the border block. The border block refers to the third phase of the task, where stimuli may or may not be bordered. The Cronbach's alpha for this task was 0.80.

**Writing composition.** *Narrative text composition*. To assess the students' capacity to create a short narrative text, a picture prompt was used. This method of examining written narrative skills in young children has been employed in previous studies [54,80,91]. Several dimensions were assessed: a) Productivity: the number of unique written words, correctly written sequences, written word fluency, and the total number of written words. All of these variables have been commonly used to measure compositional fluency and productivity in writing in previous research (e.g., [6,18,91,92]). b) Quality: the story structure was coded using the following elements: title, opening, characters, main event, problem, emotion, resolution and closure; written sentence fluency, percentage of correctly written words, and number of causal and temporal linguistic connectives [93,94]. c) Syntactic complexity: number of written T-units and grammatical structure [8]. T-units were calculated as syntactic units consisting of a main clause with all associated subordinate clauses and phrases. Depending on the sentence structure, a simple sentence counts as 1 T-unit, a compound sentence counts as 2 T-units, and a complex/subordinate sentence also counts as 1 T-unit. A score of 1 was assigned for each T-unit. The grammatical structure score was determined using a weighted index approach, where different types of sentences were assigned varying weights: simple sentences were scored

1, compound sentences were weighted 2, complex sentences received a weight of 3, and compound-complex sentences were weighted 4. This scoring method reflected the increasing complexity of sentence structures observed in children's language production. For example, a simple sentence like "The dog barks" contained one independent clause, while a compound sentence such as "The dog barks, and the cat meows" consisted of two independent clauses. A complex sentence like "Although the dog barks, the cat sleeps" included one independent clause and a dependent clause, and a compound-complex sentence like "Although the dog barks, the cat sleeps, and the bird chirps" had two independent clauses and one dependent clause. This system ensured that students using more complex grammatical structures received higher scores. It was created based on Granowsky and Botel's [95] work but was modified to assign 1 point to a simple sentence and 4 points to a compound-complex sentence, acknowledging the progression from basic to more advanced structures in young students' writing [96].

The average intraclass correlation coefficient (ICC) for productivity measures was .980, with 95% confidence intervals ranging from .97 to .98 ($F(118, 118) = 44.0$, $p < .001$); for quality measures, it was .91, with 95% confidence intervals ranging from .87 to .93 ($F(118, 118) = 11.0$, $p < .001$); and for syntactic complexity, it was .79, with 95% confidence intervals ranging from .71 to .84 ($F(118, 118) = 4.7$, $p < .001$).

## Procedure

This study followed the guidelines of the Research Ethics Committee at Universidad de La Laguna, which included the submission of an informed consent model to the parents of participating students. Prior to commencing the study, approval was obtained from the schools and families involved. The principal investigator provided the informed consent model, detailing the research objectives and ensuring anonymity and confidentiality, with the provision for participants to withdraw at any time. Written consent was obtained from the directors of the participating schools, who facilitated the distribution of the informed consent forms to families and collected them upon completion. Data collection in kindergarten commenced in early May 2022 and concluded by mid-month. For first grade, data collection took place in early May 2023 and also concluded by mid-month. Eighteen examiners were trained in task administration via an app designed to assess transcription skills, oral language skills, and executive functions among kindergarten students. The software was developed in Unity 2019.4.20f1, utilizing the Android platform support package. It is compatible with.NET 4.x API and requires at least Android 4.4 KitKat API for operation on mobile devices or tablets. Writing tasks in the software necessitate a Samsung tablet equipped with a stylus. Furthermore, the application requires microphone access for oral tasks. Optionally, the program supports data transmission to a server for result organization and download, developed using Django web framework version 4.2. The assessments conducted in kindergarten spanned three separate sessions across different days. Task order was counterbalanced among participants, who were allocated into three groups. Training encompassed task administration and assessment of both oral and written narrative texts. Recorded oral narratives were captured using the app and transcribed for subsequent analysis. An Olympus DM-620 digital voice recorder served as a backup in case of recording issues. A second session focused on evaluating and scoring recorded oral and written narratives, with guidance provided on entering assessment data into an Excel template. Training included using writing protocols as examples. Each examiner received a manual with correction criteria to ensure objectivity. Consistency among examiners was ensured by using the same correction protocols; results were reviewed collectively. Disagreements prompted revisiting correction criteria, iteratively applied to multiple examples. Tests were administered

in spring, coinciding with the end of the school year, during school hours, with students informed of study objectives and confidentiality. Tasks assessing written narration at the end of first grade were administered using paper and pencil.

## Data analysis

Data were analyzed using structural equation modelling (SEM) with the "Lavaan" package [97] in the R statistical software [98], assisted by ULLRToolbox [99]. Latent variables were constructed to represent exogenous constructs such as transcription, oral language, and executive functions, while productivity, quality, and syntactic complexity were treated as endogenous variables. The maximum likelihood robust (MLR) method was employed for all estimated models. To address the research objectives, SEM models were developed and evaluated using multiple indices, including chi-square statistics, the comparative fit index (CFI), the Tucker–Lewis index (TLI), the root mean square error of approximation (RMSEA), and standardized root mean square residuals (SRMRs). The acceptable model fit criteria included RMSEA values below .08, CFI and TLI values equal to or greater than .95, and SRMR values equal to or less than .05, following recommendations by Hu and Bentler [100]. Additionally, CFI and TLI values exceeding .90 were considered acceptable [101].

## Results

Table 1 presents descriptive statistics, including means, standard deviations, skewness, and kurtosis of the measures. Although some absolute values of skewness and kurtosis exceeded the thresholds of 3.0 and 10.0 recommended by Kline [101], we employed the Robust Maximum Likelihood (MLR) estimation method, which corrects for non-normality by adjusting standard errors and model fit indices. This method is particularly useful when data do not follow multivariate normal distributions, as is the case here, and is widely accepted in the structural equation modeling literature. Thus, while some variables showed high levels of skewness and kurtosis, the use of MLR mitigated this impact on parameter estimates and model fit evaluation.

The correlation matrix demonstrated a variety of interrelationships among variables, underscoring the need for additional examination using advanced analytical techniques such as SEM (Table 2).

The measurement model focuses on three core exogenous latent variables—executive functions, transcription skills, and oral narrative competence—crucial for understanding the development of children's writing in Spanish, a shallow orthography. Executive functions include cognitive processes like attention, working memory (e.g., digit span backward and oral cloze tasks), inhibitory control, and cognitive flexibility, hypothesized to strongly influence narrative quality. Transcription skills, measured through phoneme isolation, letter copying, name writing, and phoneme segmentation, are expected to play a lesser role in predicting productivity, quality, and syntactic complexity due to the shallow nature of Spanish orthography. Oral narrative competence, measured through narrative structure and verbal fluency indicators, is anticipated to significantly shape written production, highlighting the importance of oral language and executive functions over transcription skills in this context. The model also defines three endogenous variables: productivity (measured by unique words, written fluency, and total written words), quality (including narrative structure, sentence fluency, and correctly written words), and syntactic complexity (measured through written T-units and grammatical structure). A diagram (Fig 1) provides a visual overview of the relationships among latent variables and their indicators, offering an integrated perspective of writing development despite separate analysis of each model.

**Table 1. Descriptive statistics for transcription, oral language, executive functions and written composition measures.**

|  | Mean | SD | Skew | Kurtosis |
|---|---|---|---|---|
| Narrative competence |  |  |  |  |
| Narrative structure | 6.50 | 3.24 | 0.38 | -0.07 |
| Unique words | 22.43 | 13.72 | 2.13 | 7.60 |
| T-Units | 4.12 | 2.71 | 1.75 | 3.43 |
| Total number of Words | 27.08 | 24.75 | 2.40 | 9.09 |
| Transcription |  |  |  |  |
| Phoneme Isolation | 12.42 | 12.66 | 0.70 | -1.03 |
| Letter copying | 3.05 | 1.84 | -0.47 | -1.23 |
| Name writing | 1.96 | 2.40 | 1.66 | 2.90 |
| Picture word writing | 40.57 | 17.44 | -0.38 | -0.94 |
| Phoneme segmentation | 11.29 | 11.88 | 2.46 | 10.02 |
| Executive Functions |  |  |  |  |
| Attention | 16.37 | 6.32 | 0.44 | 0.23 |
| Digit spam backward | 4.03 | 2.31 | -0.26 | -0.41 |
| Oral cloze task | 1.46 | 1.19 | -0.06 | -1.17 |
| Inhibitory control | 0.65 | 0.22 | -0.43 | 0.60 |
| Cognitive flexibility | 0.20 | 0.09 | 1.97 | 8.27 |
| Productivity |  |  |  |  |
| Unique words written | 8.92 | 4.66 | 0.08 | 0.00 |
| Correctly written sequences | 8.82 | 5.94 | 0.52 | 0.35 |
| Written Word fluency | 2.67 | 1.80 | 0.96 | 1.68 |
| Total number of written words | 18.54 | 12.61 | 1.00 | 1.03 |
| Quality |  |  |  |  |
| Written narrative structure | 7.22 | 3.76 | -0.05 | -0.31 |
| Written sentence fluency | 1.35 | 0.87 | 1.17 | 1.84 |
| Percentage of correctly written words | 73.47 | 17.28 | -1.22 | 2.50 |
| Number of written casual connectors | 0.20 | 0.55 | 2.96 | 8.53 |
| Syntactic Complexity |  |  |  |  |
| Written T-units | 1.95 | 1.02 | 0.41 | 0.39 |
| Grammatical structure | 3.87 | 3.10 | 1.39 | 2.19 |

## Model 1: Text productivity

The estimated model showed good internal consistency, with an Omega coefficient of 0.809. For executive functions, its indicators attention (0.53), digit spam backward (0.63), oral cloze task (0.44), inhibitory control (0.56), and cognitive flexibility (0.20) all exhibited significant standardized factor loadings. For transcription skills, the indicators phoneme isolation (0.78), letter copying (0.39), name writing (0.57), picture word writing (0.61), and phoneme segmentation (0.60) showed significant standardized factor loadings. With respect to narrative competence, narrative structure (0.51), unique word written (0.96), T-units (0.91), and total number of written words (1.00) were also robust indicators of the latent construct. Moreover, the latent variable productivity demonstrated significant paths to its indicators unique words written (0.95), correctly written sequences (0.96), written word fluency (0.90), and total number of written words (0.70). As shown in Table 3, these significant factor loadings suggest substantial connections between observable and latent variables.

The model estimates indicate a good fit to the observed data. Notably, the chi-square statistic was significant ($\chi^2$ = 202.164, $p < 0.001$), suggesting a discrepancy between the proposed

**Table 2. Intercorrelations for scores on transcription, oral language, executive functions and written composition measures.**

| | 1 | 2 | 3 | 4 | 5 | 6 | 7 | 8 | 9 | 10 | 11 | 12 | 13 | 14 | 15 | 16 | 17 | 18 | 19 | 20 | 21 | 22 | 23 |
|---|---|---|---|---|---|---|---|---|---|---|---|---|---|---|---|---|---|---|---|---|---|---|---|
| 1. NS | | | | | | | | | | | | | | | | | | | | | | | |
| 2. UW | .494*** | | | | | | | | | | | | | | | | | | | | | | |
| 3. TU | .546*** | .781*** | | | | | | | | | | | | | | | | | | | | | |
| 4. TNW | .510*** | .939*** | .886*** | | | | | | | | | | | | | | | | | | | | |
| 5. PI | .096 | -.047 | -.024 | -.030 | | | | | | | | | | | | | | | | | | | |
| 6. LC | -.086 | .007 | -.029 | -.043 | .248*** | | | | | | | | | | | | | | | | | | |
| 7. NW | -.025 | .039 | -.010 | .004 | .387*** | .223*** | | | | | | | | | | | | | | | | | |
| 8. WPIC | .084 | -.013 | .033 | -.019 | .476*** | .126* | .428*** | | | | | | | | | | | | | | | | |
| 9. PS | .036 | -.080 | -.040 | -.069 | .486*** | .166** | .216*** | .398*** | | | | | | | | | | | | | | | |
| 10. AT | .063 | -.094 | -.079 | -.121* | .252** | .155** | .278*** | .268*** | .254*** | | | | | | | | | | | | | | |
| 11. DSB | .057 | .006 | -.010 | -.047 | .352*** | .244*** | .318*** | .303*** | .210*** | .345*** | | | | | | | | | | | | | |
| 12. OCT | .084 | -.029 | -.008 | -.033 | .249*** | .265*** | .228*** | .208*** | .221*** | .238*** | .362*** | | | | | | | | | | | | |
| 13. IC | -.018 | -.027 | -.063 | -.060 | .212*** | .212*** | .247*** | .149* | .199** | .346*** | .347*** | .264*** | | | | | | | | | | | |
| 14. CF | -.083 | -.001 | -.033 | .001 | .039 | .135* | .117* | .098 | .065 | .138* | .116* | .011 | .225*** | | | | | | | | | | |
| 15. UWW | .106 | .010 | -.029 | -.032 | .353*** | .224*** | .315*** | .331*** | .192** | .331*** | .338*** | .239*** | .263*** | .140* | | | | | | | | | |
| 16. CWS | .115 | .007 | -.028 | -.011 | .411*** | .218*** | .366*** | .368*** | .211** | .328*** | .324*** | .244*** | .259*** | .115* | .911*** | | | | | | | | |
| 17. WWF | .070 | -.027 | -.082 | -.050 | .403*** | .223*** | .348*** | .399*** | .233*** | .377*** | .377*** | .236*** | .306*** | .117* | .842*** | .832*** | | | | | | | |
| 18. TNWW | .197*** | .060 | .054 | .034 | .342*** | .147* | .399*** | .356*** | .205** | .292*** | .359*** | .271*** | .228*** | .071 | .665*** | .671*** | .586*** | | | | | | |
| 19. WNS | .199*** | .093 | .119* | .093 | .279*** | .200*** | .311*** | .248*** | .184** | .282*** | .319*** | .233*** | .239*** | .089 | .573*** | .553*** | .459*** | .702*** | | | | | |
| 20. WSF | .044 | .005 | .001 | .003 | .239*** | .105 | .161** | .203*** | .171** | .193*** | .157** | .111 | .158** | .088 | .457*** | .400*** | .441*** | .377*** | .280*** | | | | |
| 21. PCW | -.013 | -.015 | -.008 | -.005 | .233*** | .183** | .211*** | .289*** | .144* | .163** | .288*** | .123* | .202*** | .071 | .209*** | .228*** | .464*** | .277*** | .206*** | .100 | | | |
| 22. NWCC | .067 | .121* | .110 | .088 | .016 | .078 | .094 | .135* | .033 | .070 | .094 | .062 | .118* | .073 | .178*** | .166** | .120* | .317*** | .363*** | .040 | .075 | | |
| 23. WTU | .087 | .082 | .054 | .094 | .217*** | .166** | .185** | .236*** | .083 | .260*** | .201*** | .184*** | .184* | .127* | .658*** | .638*** | .599*** | .449*** | .429*** | .596*** | .182** | .057 | |
| 24. GS | .087 | .078 | .075 | .080 | .293*** | .142* | .277*** | .241*** | .229*** | .250*** | .253*** | .231*** | .199*** | .102 | .472*** | .470*** | .414*** | .734*** | .572*** | .584*** | .203*** | .277*** | .496*** |

Note

*$p < .05$

**$p < .01$

***$p < .001$. NS = narrative structure; UW = unique words; TU = T-units; TNW = total number of words; PI = phoneme isolation: LC = letter copying; NW = name writing; WPI = picture word writing; PS = phoneme segmentation; AT = attention; DSB = digit spam backward; OCT = oral cloze task; CF = cognitive flexibility; UWW = unique words written; CWS = correctly written sequences; WWF = written word fluency; TNWW = total number of written words; WNS = written narrative structure; WSF = written sentence fluency; PCW = percentage of correctly written words; NWC = number of written casual connectors; WTU = written T-units; GS = grammatical structure.

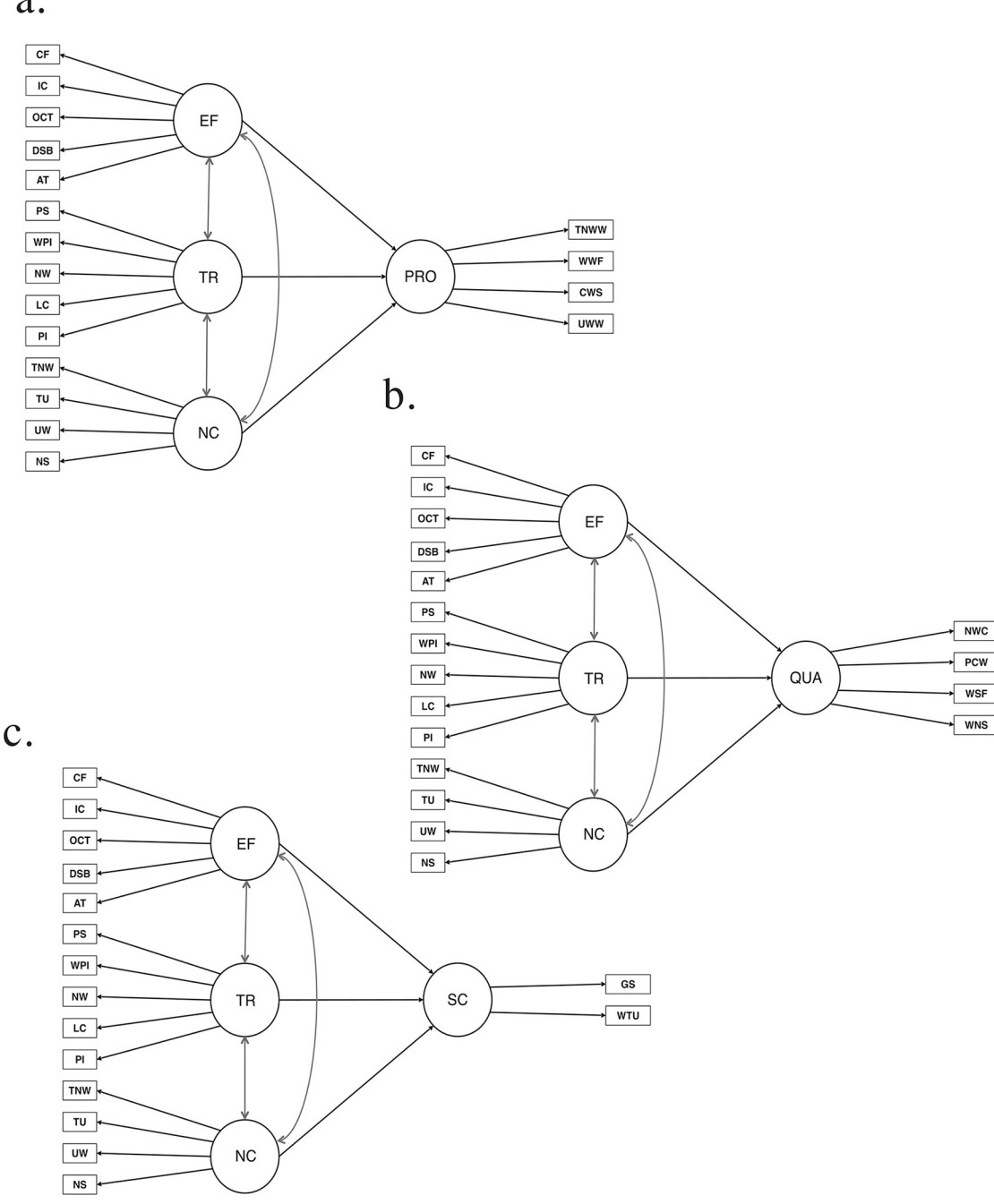

**Fig 1. Three models of the effects of transcription skills, oral language skills, and executive functions on the writing dimensions of productivity, quality, and syntactic complexity are presented.** Black lines represent predictive paths, and grey lines represent covariances. EF = Executive functions; CF = cognitive flexibility; IC = inhibitory control; OCT = oral cloze task; DSB = digit spam backward; AT = attention; TR = transcription skills; PS = phoneme segmentation, WPI = picture word writing; NW = name writing; LC = letter copying; PI = phoneme isolation; NC = narrative competence; TNW = total number of words; TU = T-units; UW = unique words written; NS = narrative structure; PRO = productivity; TNWW = total number of written words; WWF = written word fluency; CWS = correctly written sequences; UWW = unique words written; QUA = quality; NWC = number of written casual connectors; PCW = percentage of correctly written words; WSF = written sentence fluency; WNS = written narrative structure; SC = syntactic complexity; GS = grammatical structure; WTU = written T-units.

**Table 3. Unstandardized and standardized path coefficients for transcription skills, narrative competence, executive functions, and productivity.**

| Path | Unstandardized | Standardized |
|---|---|---|
| Narrative Competence | | |
| Narrative Competence→ Productivity | 0.02 | 0.00 |
| Narrative structure | 1.00[+] | 0.51*** |
| Unique words | 7.23 | 0.96*** |
| T-Units | 1.52 | 0.91*** |
| Total number of Words | 14.44 | 1.00*** |
| Transcription skills | | |
| Transcription skills→ Productivity | 0.16 | 0.38*** |
| Phoneme Isolation | 1.00[+] | 0.78*** |
| Letter copying | 0.07 | 0.39*** |
| Name writing | 0.14 | 0.57*** |
| Picture word writing | 1.00 | 0.61*** |
| Phoneme segmentation | 0.66 | 0.60*** |
| Executive function | | |
| Executive function → Productivity | 0.34 | 0.271* |
| Attention | 1.00[+] | 0.53*** |
| Digit spam backward | 0.40 | 0.63*** |
| Oral cloze task | 0.14 | 0.44*** |
| Inhibitory control | 0.04 | 0.56*** |
| Cognitive flexibility | 0.00 | 0.20* |
| Productivity | | |
| Unique words written | 1.00[+] | 0.95*** |
| Correctly written sequences | 1.35 | 0.96*** |
| Written Word fluency | 0.37 | 0.90*** |
| Total number of written words | 1.98 | 0.70*** |

[+]Fixed parameter

*$p < .05$

**$p < .01$

***$p < .001$.

model and the observed data. However, this statistic tends to be sensitive to sample size, so it is important to consider other fit indices [102]. The NFI showed a value of 0.908, indicating that the model explained 90.8% of the variance and covariance of the observed data. Similarly, the NNFI had a value of 0.958, while the CFI and TLI had values of 0.964. These indices surpassed the recommended threshold of 0.95, suggesting a good fit of the model. The RMSEA was 0.054, with a 90% confidence interval between 0.040 and 0.069, indicating an acceptable model fit. Additionally, a scaled version of the chi-square test (scaled) demonstrated a good fit ($\chi^2$ = 212.998, $p < 0.001$; RMSEA = 0.057. In summary, these results suggest that the proposed model fit well with the observed data, explaining substantial variance and covariance in the latent and observed variables.

In terms of structural paths, transcription skills significantly predicted text productivity with a standardized path coefficient of 0.381 ($p < 0.01$), indicating a positive influence of transcription on productivity. Similarly, EF also significantly predicted text productivity with a standardized path coefficient of 0.271 ($p < 0.05$), suggesting a positive influence of executive functions on productivity. However, the path from the narrative competence to the text productivity ($\beta = .009$) ($p = .90$) was not statistically significant (Fig 2).

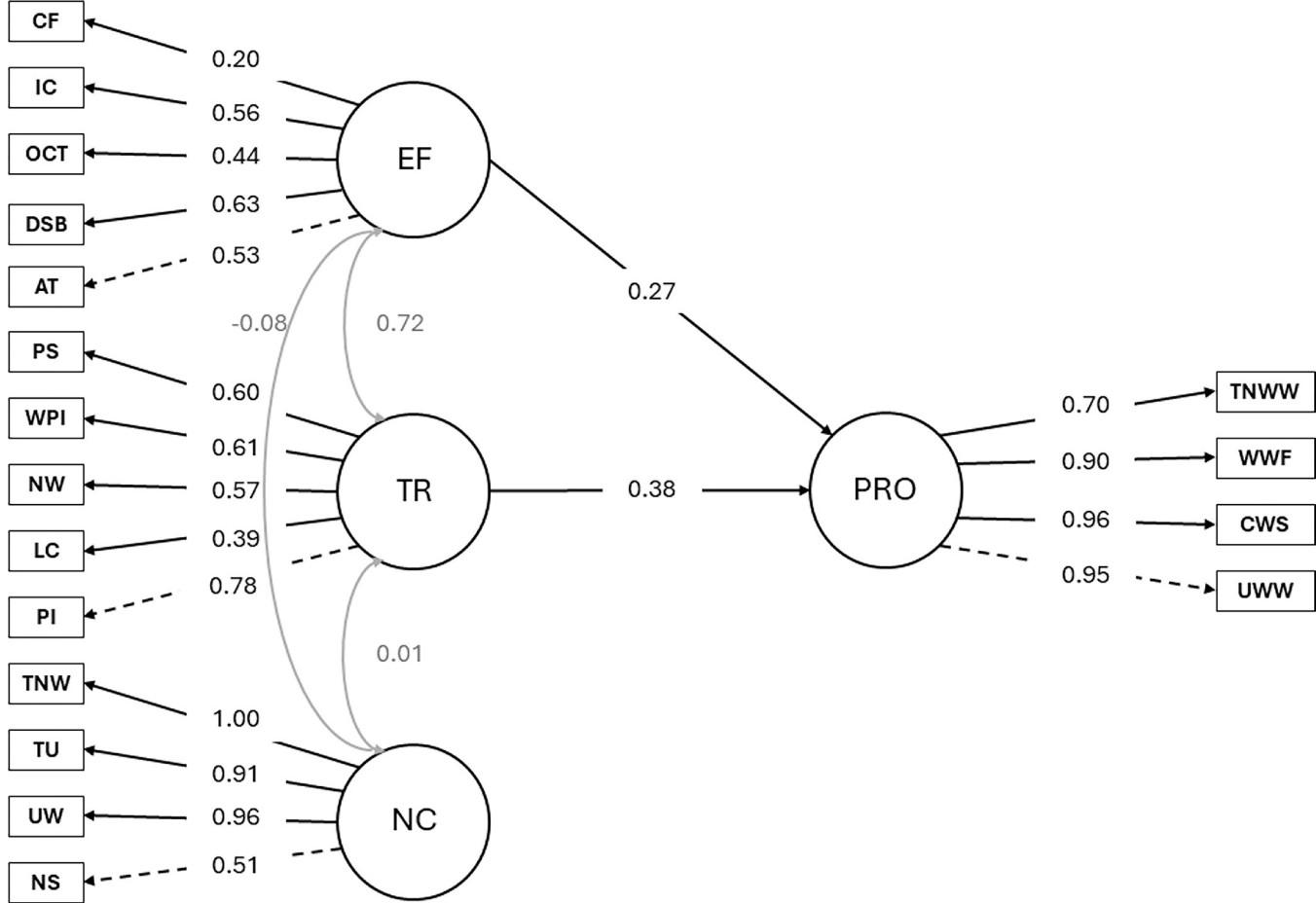

**Fig 2. This graph shows a structural model of the relationship between executive functions, transcription skills, narrative competence, and productivity (Model 1).** Circles represent factors (i.e., latent variables), rectangles represent indicators (i.e., observed variables), black arrows represent direct paths, and gray arrows represent covariances. EF = Executive functions; CF = cognitive flexibility; IC = inhibitory control; OCT = oral cloze task; DSB = digit spam backward; AT = attention; TR = transcription skills; PS = phoneme segmentation, WPI = picture word writing; NW = name writing; LC = letter copying; PI = phoneme isolation; NC = narrative competence; TNW = total number of words; TU = T-units; UW = unique words written; NS = narrative structure; PRO = productivity; TNWW = total number of written words; WWF = written word fluency; CWS = correctly written sequences; UWW = unique words written.

## Model 2: Text quality

As shown in Table 4, the measurement model produced significant standardized factor loadings for all latent variables and their respective indicators. A total omega coefficient of 0.887 was observed, indicating good overall reliability of the model.

For narrative competence, strong factor loadings were observed for its indicators, namely, narrative structure (0.51), unique words (0.96), T-units (0.92), and total number of words (1.00). For transcription, significant factor loadings were found for phoneme isolation (0.78), letter copying (0.43), naming writing (0.56), picture word writing (0.59), and phoneme segmentation (0.61). Likewise, for EF, significant factor loadings were observed for attention (0.52), digit spam backward (0.60), oral cloze task (0.43), inhibitory control (0.57), and cognitive flexibility (0.21). Notably, the factor loading for cognitive flexibility was lower than other indicators, though it was still statistically significant. Finally, for text quality, strong factor loadings were evident for written narrative structure (0.79), written sentence fluency (0.17), percentage of correctly written words (0.35), and number of written causal connectors (0.45). The

**Table 4. Unstandardized and standardized path coefficients for transcription skills, narrative competence, executive functions, and quality.**

| Path | Unstandardized | Standardized |
|---|---|---|
| Narrative Competence | | |
| Narrative Competence→ Quality | 0.29 | 0.18* |
| Narrative structure | 1.00+ | 0.51*** |
| Unique words | 7.18 | 0.96*** |
| T-Units | 1.49 | 0.92*** |
| Total number of words | 14.37 | 1.00*** |
| Transcription skills | | |
| Transcription skills→ Quality | 0.09 | 0.34* |
| Phoneme isolation | 1.00+ | 0.78*** |
| Letter copying | 0.07 | 0.43*** |
| Name writing | 0.14 | 0.56*** |
| Picture word writing | 1.01 | 0.59*** |
| Phoneme segmentation | 0.66 | 0.61*** |
| Executive function | | |
| Executive function → Quality | 0.23 | 0.29 |
| Attention | 1.00+ | 0.52*** |
| Digit spam backward | 0.40 | 0.60*** |
| Oral cloze task | 0.14 | 0.43*** |
| Inhibitory control | 0.04 | 0.57*** |
| Cognitive flexibility | 0.00 | 0.21* |
| Quality | | |
| Written narrative structure | 1.00+ | 0.79*** |
| Written sentence fluency | 0.05 | 0.17 |
| Percentage of correctly written words | 2.19 | 0.35*** |
| Number of written casual connectors | 1.98 | 0.45*** |

+Fixed parameter

*$p < .05$

**$p < .01$

***$p < .001$.

significant factor loadings suggest robust relationships between the latent variables and their indicators. The model fit results obtained through the MLR method indicate a satisfactory overall fit. The chi-square statistic was significant ($\chi^2 = 184.655$, $df = 129$, $p < 0.001$), suggesting some deviation between the proposed model and the observed data. However, given the sensitivity of the chi-square statistic to sample size, additional fit indices were considered. The NFI was 0.871, while the NNFI was 0.948. Furthermore, the CFI and TLI were 0.956 and 0.948, respectively. Despite the TLI slightly below the recommended threshold of 0.95, both indices indicate a good model fit. The MFI was 0.862. Additionally, the RMSEA was 0.048, with a 90% confidence interval between 0.031 and 0.063, suggesting an acceptable model fit. A scaled version of the chi-square statistic yielded a value of 195.723 ($df = 129$, $p < 0.001$), with an RMSEA of 0.051 and a 90% confidence interval between 0.036 and 0.065. Regarding the structural model, the relationships between text quality and both transcription ($\beta = 0.344$, $p < 0.05$) and narrative competence ($\beta = 0.184$, $p < 0.05$) were found to be statistically significant. However, the association between text quality and EF did not reach statistical significance ($\beta = 0.291$, $p = 0.156$) (Fig 3).

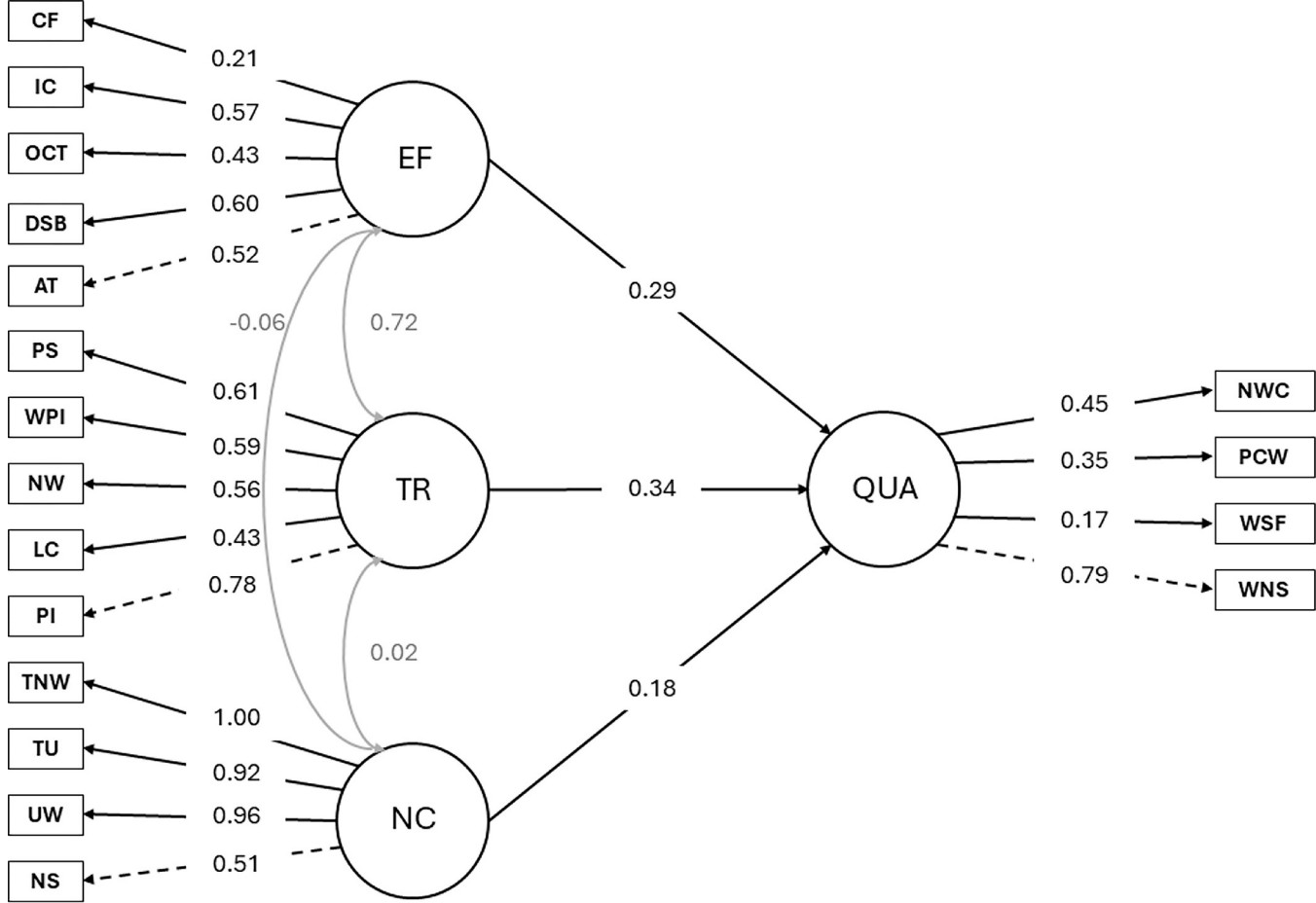

**Fig 3. This graph shows a structural model of the relationship between executive functions, transcription skills, narrative competence, and quality (Model 2).** Circles represent factors (i.e., latent variables), rectangles represent indicators (i.e., observed variables), black arrows represent direct paths, and gray arrows represent covariances. EF = Executive functions; CF = cognitive flexibility; IC = inhibitory control; OCT = oral cloze task; DSB = digit spam backward; AT = attention; TR = transcription skills; PS = phoneme segmentation, WPI = picture word writing; NW = name writing; LC = letter copying; PI = phoneme isolation; NC = narrative competence; TNW = total number of words; TU = T-units; UW = unique words written; NS = narrative structure; QUA = quality; NWC = number of written casual connectors; PCW = percentage of correctly written words; WSF = written sentence fluency; WNS = written narrative structure.

## Model 3: Syntactic complexity

Table 5 shows the unstandardized and standardized path coefficients for transcription skills, narrative competence, executive functions, and syntactic complexity. The Omega total coefficient was 0.833, further supporting the internal consistency of the measured variables.

The measurement model produced significant standardized factor loadings for all latent variables and their respective indicators. For narrative competence, significant standardized factor loadings were observed for narrative structure (0.52), unique words (0.96), T-units (0.91), and total number of words (1.00), indicating strong associations between these indicators and the latent variable. Transcription skills demonstrated robust standardized factor loadings for phoneme isolation (0.79), letter copying (0.38), name writing (0.54), picture word writing (0.61), and phoneme segmentation (0.61), highlighting their substantial contributions to the transcription skills construct. Executive function exhibited significant standardized factor loadings for attention (0.55), digit span backward (0.58), oral cloze task (0.46), inhibitory control (0.56), and cognitive flexibility (0.20), indicating their influence on the executive

**Table 5. Unstandardized and standardized path coefficients for transcription skills, narrative competence, executive functions, and syntactic complexity.**

| Path | Unstandardized | Standardized |
|---|---|---|
| Narrative Competence | | |
| Narrative Competence→ Syntactic Complexity | 0.07 | 0.191* |
| Narrative structure | 1.00+ | 0.52*** |
| Unique words | 7.29 | 0.96*** |
| T-Units | 1.53 | 0.91*** |
| Total number of Words | 14.61 | 1.00*** |
| Transcription skills | | |
| Transcription skills→ Syntactic Complexity | 0.01 | 0.20 |
| Phoneme Isolation | 1.00+ | 0.79*** |
| Letter copying | 0.07 | 0.38*** |
| Name writing | 0.13 | 0.54*** |
| Picture word writing | 0.99 | 0.61*** |
| Phoneme segmentation | 0.68 | 0.61*** |
| Executive function | | |
| Executive function → Syntactic Complexity | 0.08 | 0.46** |
| Attention | 1.00+ | 0.55*** |
| Digit spam backward | 0.36 | 0.58*** |
| Oral cloze task | 0.14 | 0.46*** |
| Inhibitory control | 0.04 | 0.56*** |
| Cognitive flexibility | 0.004 | 0.20* |
| Syntactic complexity | | |
| Grammatical structure | 1.00+ | 0.71*** |
| Written T-units | 3.62 | 0.64*** |

+Fixed parameter

*$p < .05$

**$p < .01$

***$p < .001$.

function latent variable. Syntactic complexity was strongly influenced by grammatical structure (0.71) and written T-units (0.64), underscoring their importance as indicators of syntactic complexity in the model.

The model fit statistics obtained through the MLR method indicate a satisfactory overall fit. The chi-square statistic was significant ($\chi^2 = 139.668$, $df = 98$, $p < 0.004$), indicating some deviation between the proposed model and the observed data. However, considering the sensitivity of the chi-square statistic to sample size, additional fit indices were evaluated. The NFI had a value of 0.897, while the NNFI was 0.959. Furthermore, the CFI and TLI were 0.966 and 0.959, respectively, surpassing the recommended threshold of 0.95, indicating a good model fit. The MFI was 0.893. Additionally, the RMSEA was 0.048, with a 90% confidence interval between 0.028 and 0.065, suggesting an acceptable model fit. A scaled version of the chi-square statistic yielded a value of 145.112 ($df = 98$, $p < 0.001$), with an RMSEA of 0.050 and a 90% confidence interval between 0.032 and 0.067. These findings indicate that the proposed model aligned well with the observed data, explaining substantial variance and covariance among the latent and observed variables. The structural model revealed several significant pathways linking latent variables. Specifically, the standardized estimates indicated that narrative competence significantly predicted syntactic complexity (β = 0.191, $p < .05$), while transcription had a beta value

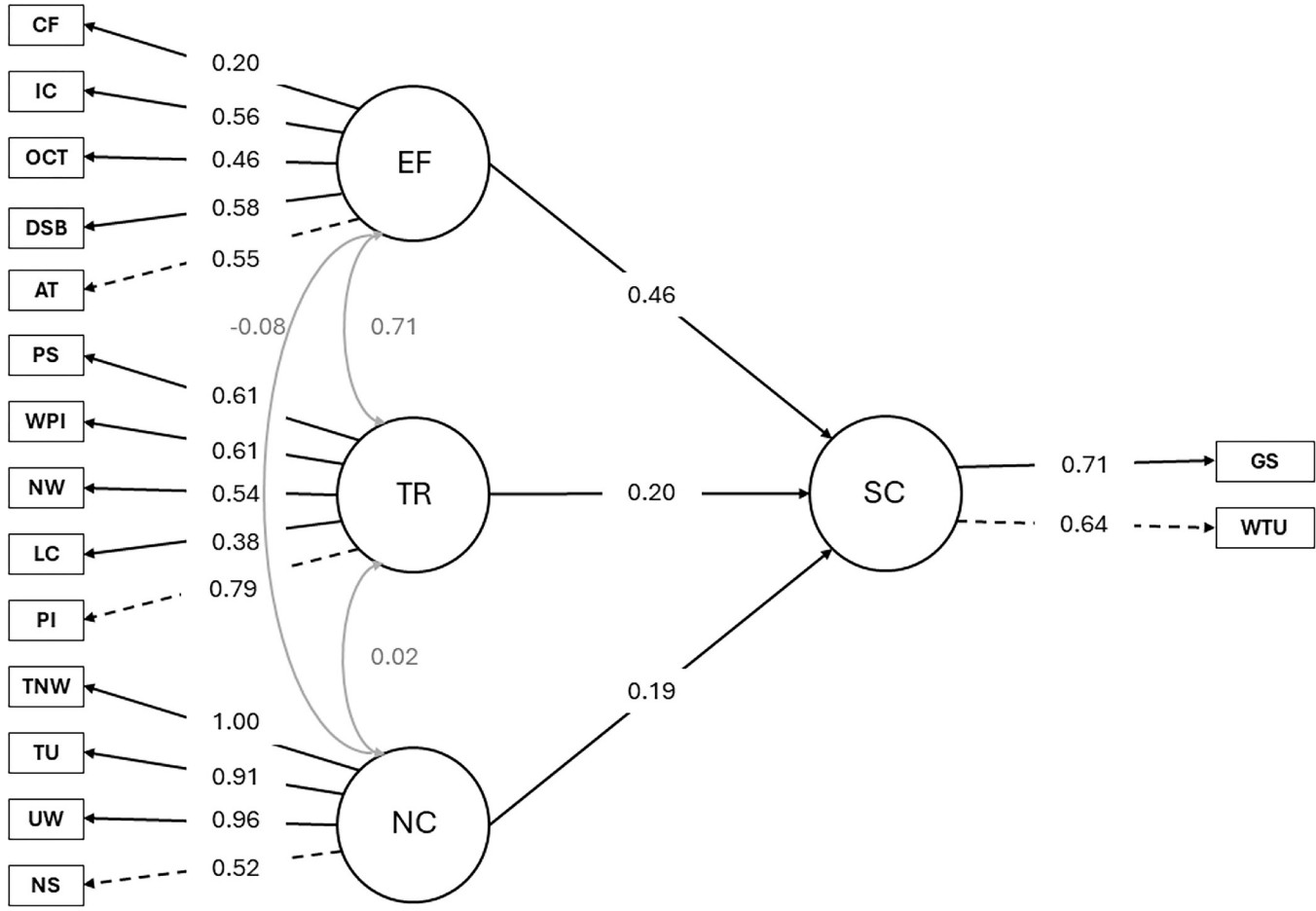

**Fig 4. Structural model of the relationship between executive functions, transcription skills, narrative competence, and syntactic complexity (Model 3).**
Circles represent factors (i.e., latent variables), rectangles represent indicators (i.e., observed variables), black arrows represent direct paths, and gray arrows represent covariances. EF = Executive functions; CF = cognitive flexibility; IC = inhibitory control; OCT = oral cloze task; DSB = digit spam backward; AT = attention; TR = transcription skills; PS = phoneme segmentation, WPI = picture word writing; NW = name writing; LC = letter copying; PI = phoneme isolation; NC = narrative competence; TNW = total number of words; TU = T-units; UW = unique words written; NS = narrative structure; SC = syntactic complexity; GS = grammatical structure; WTU = written T-units.

of 0.202, but this effect was not statistically significant ($p = 0.261$). However, executive functioning demonstrated a stronger and more significant association with syntactic complexity ($\beta = 0.460$, $p < .01$), suggesting a robust influence of executve functioning on syntactic complexity (Fig 4).

For all three models, we observed robust covariance between the latent variables of transcription skills and executive function constructs, indicating that elevated scores in executive functions often align with higher scores in transcription.

## Discussion

The present study examines the longitudinal contributions of transcription skills, oral language abilities, and executive functions to various dimensions of written production in first-grade students starting from kindergarten. We anticipated that in a shallow orthographic system such as Spanish, transcription skills might have a lesser impact than oral language skills and executive functions on dimensions of written narratives such as productivity, quality, and

syntactic complexity. However, our findings revealed a nuanced picture. Transcription skills and executive functions contributed significantly to productivity, whereas transcription skills and oral language abilities were influential in determining text quality. Furthermore, syntactic complexity was influenced by oral language skills and executive functions. Thus, our results partially confirmed our initial expectations. While transcription skills played a significant role in certain dimensions of written production, the combined influence of oral language skills and executive functions also emerged as pivotal factors in shaping written expression in Spanish.

Consistent with the simple view of writing and its extension, the not-so-simple view of writing, our findings reaffirm the notion that transcription skills play a crucial role in converting ideas into written symbols, thereby facilitating the text generation process [10,11]. Handwriting fluency and spelling proficiency enable children to effectively translate their thoughts into written language, contributing to both the quantity and quality of their written narratives [73,103]. This result aligns with findings from previous studies conducted in Spanish, which showed that spelling was a significant predictor of sentence-level writing fluency and the quality of narrative writing. This relationship was observed in children up to Grade 2 [60], particularly when graphonomic measures were employed [61]. Principio del FormularioThe strong association between transcription skills and productivity suggests that proficient handwriting and spelling are essential for generating a sufficient volume of written text. Children with greater transcription skills may produce more words and sentences due to their ability to efficiently transcribe ideas into print. This finding aligns with previous research demonstrating the positive impact of handwriting fluency and spelling accuracy on writing productivity [18,22]. Moreover, we found that transcription skills significantly predicted writing quality, highlighting their role in shaping the coherence, organization, and accuracy of written narratives. In other words, children with stronger handwriting and spelling abilities are more likely to produce texts that exhibit higher levels of linguistic proficiency and syntactic accuracy. This finding underscores the importance of transcription skills in facilitating the language-to-text transformation process, particularly in languages with shallow orthographies where spelling accuracy may receive less emphasis [5,22].

Contrary to our expectations, while the direct influence of transcription skills on syntactic complexity was not statistically significant in our study, executive functions were found to play a significant role. This suggests that in early childhood, children rely on executive functions to construct sentences and convey their ideas in written narratives. In transparent orthographies like Spanish and Icelandic, transcription skills such as spelling and handwriting typically become automatized at younger ages, allowing cognitive resources to be redirected towards more complex aspects of writing, such as syntactic complexity. As our study and that of Oddsdóttir et al. [59] demonstrate, executive functions—particularly self-regulation—may therefore play a more central role in syntactic complexity among younger children writing in transparent orthographies.

In contrast, most studies examining the relationship between executive functions and writing have been conducted with older children in opaque orthographies, where transcription skills are acquired later and remain a more prominent factor in writing development. For example, Drijbooms et al. [56] found correlations between executive functions and writing in students aged 8–15 years, while Cordeiro et al. [13] highlighted the role of working memory and planning in predicting text quality in second graders. Therefore, due to the slower automatization process in opaque orthographies, transcription skills remain a significant contributor in older children in these studies.

Despite this insignificant finding, the role of transcription skills in supporting syntactic development should not be overlooked. Handwriting and spelling are commonly regarded as

foundational skills necessary for syntactic experimentation and elaboration in written narratives [53,103]. However, in transparent orthographies, these skills may become automated earlier, allowing executive functions to take on a more prominent role in driving syntactic complexity. Interestingly, Limpo and Alves [22] found that children's ability to plan, review, and revise text predicted text quality in fourth graders, suggesting that the influence of executive functions on writing quality becomes increasingly important as children develop. Our study did not find evidence supporting the expected relationship between transcription skills and syntactic complexity, reinforcing the need to further explore how the transparency of an orthography affects the dynamics between transcription and executive functions in writing development.

The findings of the study also revealed that oral narrative competence significantly contributed to both writing quality and syntactic complexity in children's written narratives. This highlights the crucial role of oral language skills in shaping various dimensions of writing, even in the context of a shallow orthography such as Spanish. Rodríguez et al. [62] demonstrated an early influence of oral language skills on the productivity and quality of narrative writing in Spanish kindergarten children. Oral language skills may influence writing quality through vocabulary knowledge and syntactic proficiency. Children with stronger oral language skills possess a richer vocabulary and a better understanding of sentence structure, which enable them to express their ideas more precisely in writing [30]. Additionally, oral language proficiency may facilitate the organization of ideas and the development of coherent story-telling, leading to higher-quality written compositions [35]. Moreover, oral language skills may contribute to syntactic complexity in children's writing [16,104]. Proficient oral language users are likely to have a deeper understanding of sentence structure and grammatical conventions. As a result, they construct more complex sentences by using a wider range of syntactic structures in their written narratives [65]. Thus, oral language proficiency serves as a foundation for syntactic development in writing, allowing children to incorporate syntactic complexity into their compositions as they become more proficient writers.

Notably, the role of oral language skills in early writing development has been a topic of debate, particularly for Grade 1 students. In studies conducted in opaque orthographies, oral language skills are not related to text production in Grade 1 due to the constraints imposed by transcription skills (e.g., [1,18,26]). In contrast, in our study, both transcription skills and oral narrative skills were found to influence the dimension of textual quality. Taken together, these findings suggest that the timing and extent of the impact of oral language skills on early writing development may be influenced by the transparency or opacity of the orthography. In a shallow orthographic context, where spelling accuracy is achieved quickly, oral language skills play a crucial role in shaping the quality of written narratives from an early stage. Our finding highlights the nuanced interplay between orthographic transparency, transcription skills, and oral language abilities in shaping early writing proficiency. In fact, this finding aligns with prior research by Rodríguez et al. [62], who observed similar results in kindergarten children, in the context of the transparent Spanish orthography. These authors suggested that transcription and oral language skills play comparable roles in determining writing quality, challenging the traditional view of transcription as a bottleneck that limits cognitive resources for higher-order writing tasks [105,106].

We also observed a significant positive influence of EFs on productivity in children's writing. The association between EFs and productivity underscores the importance of considering cognitive processes beyond transcription skills and oral language abilities in writing development. While transcription skills contribute to productivity by enabling efficient language-to-text transformation, EFs provide the cognitive framework necessary for organizing and executing writing tasks effectively [46]. Contrary to our expectations, EFs did not exhibit a significant

direct influence on writing quality in our study. One possible explanation for this is that in early childhood, oral narrative competence is more strongly associated with the overall quality of written narratives than executive functions, a relationship that can be attributed to several factors. Both oral and written narratives share a common foundation of linguistic skills, such as vocabulary, grammar, and narrative structure. Children who are adept at oral storytelling possess a solid understanding of how to construct coherent and structured stories, which directly translates into their writing. Oral narratives are often practiced in social contexts where children receive immediate feedback and support. This practice establishes a clear structure that children can then transfer to their writing, thereby improving the quality of their written narratives [107]. However, while narrative competence is more strongly associated with the overall quality of written narratives, EFs have a significant influence on the syntactic complexity of these texts, including grammatical structures and T-units. EFs, such as working memory, cognitive flexibility, and inhibitory control, are crucial for managing the multiple cognitive processes involved in constructing complex sentences and maintaining grammatical accuracy [16]. These functions enable children to plan and organize their thoughts, manipulate linguistic information, and sustain attention on intricate language tasks. Consequently, EFs contribute more directly to the syntactic sophistication of written narratives, whereas narrative competence primarily enhances the coherence and richness of the content [16,107].

Our study highlights the association between EFs and syntactic complexity in young Spanish children. This finding is similar to that observed in later developmental stages, as demonstrated by Drijbooms et al. [16] in fourth and sixth grades. By identifying this relationship in younger children, our study increases our understanding of the developmental trajectory of writing skills. Furthermore, despite the reduced emphasis on spelling accuracy in languages with shallow orthographies [5,30], challenges persist due to the intricacies of grammar and morphology [67,68]. These complexities can present significant obstacles for young writers in narrative construction. In such contexts, the critical role of executive functions in supporting narrative expression, especially in contexts where spelling accuracy may not be the primary focus. Therefore, our study sheds light on how linguistic characteristics, such as orthographic transparency, influence the cognitive processes involved in narrative writing development, emphasizing the multifaceted nature of early writing proficiency.

Notably, the pronounced covariance between the executive functions and transcription constructs indicates that elevated scores in executive functions were associated with higher scores in transcription. This result is consistent with the not-so-simple view of writing [10,11], highlighting the interrelationship between these processes in the initial stage of knowledge-telling. Executive functions facilitate transcription acquisition, while automatized transcription frees cognitive resources for executive functions to support higher-order composition processes.

An interesting finding that emerged in our study is the low loading of cognitive flexibility on the construct of executive functioning, as represented by the latent variable. This low loading may be explained by several factors. First, while executive functions are typically viewed as a unitary construct in early childhood [38], certain subcomponents, such as cognitive flexibility, may develop at a slower rate compared to inhibitory control or working memory [108]. This developmental delay could lead to lower loadings in younger children. Second, the tasks used to measure cognitive flexibility may not fully capture age-appropriate manifestations of this ability, potentially affecting performance and its representation in the model [109]. Finally, despite the unitary nature of executive functions in early childhood, cognitive flexibility may begin to differentiate slightly from other components, suggesting emerging distinctions in the development of executive functions [108].

## Educational implications

The findings of our study, which examined Spanish, a shallow orthography, have significant educational implications. Our results underscore the importance of transcription skills, oral language abilities, and executive functions in early writing development. While transcription skills play a crucial role in generating written text, oral language skills and executive functions also have a substantial influence on different dimensions of writing. As a result, writing instruction programs should address not only writing practice but also oral and executive skills to enhance students' writing quality and syntactic complexity.

Our findings support the need to integrate teaching practices that foster the development of transcription skills from the early stages of learning to write. Since transcription skills are closely related to productivity and writing quality, educators should incorporate writing activities that encourage handwriting fluency and spelling accuracy, which in turn increase text production and quality. Furthermore, the significant influence of oral language skills on writing quality and syntactic complexity highlights the need of promoting oral language development in the classroom. Educational programs should include activities that promote vocabulary development, grammatical understanding, and oral storytelling, as these skills enrich students' writing and enhance their ability to express ideas coherently and sophisticatedly. Finally, educators can implement activities that specifically target executive functions, such as storytelling activities that require students to sequence events in order, problem-solving tasks, and activities to improve attention and self-regulation. By providing a variety of engaging activities that target transcription skills, oral language skills, and executive functions, educators can create a learning environment that promotes the development of proficient writing skills.

## Limitations and future research

The findings of our study must be considered in the context of its limitations. One notable constraint was the relatively small sample size, which may have hindered our ability to detect smaller effects and reduced the generalizability of the findings. Additionally, the study was conducted in a specific geographical region (the Canary Islands), which may limit the generalizability of the results to other regions of Spain or beyond. Future research should aim to include samples from different regions to obtain more robust conclusions and account for potential regional differences in educational practices or student characteristics.

Another limitation is the absence of a non-verbal intelligence measure, which is often included in research related to executive functions. We also did not assess other linguistic skills such as orthographic knowledge and morphological awareness. These measures are particularly relevant in the context of a shallow orthography, as they may play a crucial role in the development of writing abilities. Acknowledging these limitations, we suggest that future research incorporate these variables to provide a more comprehensive understanding of the cognitive and linguistic factors influencing writing performance.

Furthermore, there is evidence suggesting that foundational skills (i.e., transcription, text generation, and executive functions) affect narrative and descriptive texts differently, as children activate their text generation skills for narrative writing earlier than for informational texts [59]. Therefore, our findings cannot be generalized to other text genres. Future studies examining the role of these skills in writing informational texts will allow us to understand whether the relationships identified in our study are universal or limited to a specific genre [4]. Furthermore, the study did not explore the potential influence of teacher characteristics or instructional practices on students' writing development. Research in this area will provide valuable insights into the role of the educational environment in shaping writing proficiency. Moving forward, future longitudinal research should address these limitations by employing

larger and more diverse samples and exploring the influence of contextual factors such as teacher characteristics, instructional practices, and technology integration on students' writing development.

## Supporting information

**S1 Table. Unstandardized and standardized path coefficients for transcription skills, narrative competence, executive functions, and productivity.**
(DOCX)

**S2 Table. Unstandardized and standardized path coefficients for transcription skills, narrative competence, executive functions, and quality.**
(DOCX)

**S3 Table. Unstandardized and standardized path coefficients for transcription skills, narrative competence, executive functions, and syntactic complexity.**
(DOCX)

**S1 Dataset. Raw data for the analysis of transcription skills, executive functions, narrative competence, and the writing dimensions of productivity, quality, and syntactic complexity.**
(XLSX)

**S1 File.**
(XLSX)

## Acknowledgments

The authors would like to thank all the participating schools, teachers, students, and families.

## Author Contributions

**Conceptualization:** Juan E. Jiménez, Becky Xi Chen.

**Formal analysis:** Jennifer Balade, Eduardo García.

**Funding acquisition:** Juan E. Jiménez.

**Methodology:** Juan E. Jiménez.

**Supervision:** Juan E. Jiménez, Becky Xi Chen.

**Writing – original draft:** Juan E. Jiménez.

**Writing – review & editing:** Jennifer Balade, Eduardo García, Becky Xi Chen.

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
