## [Decision Letter · Decision Letter 0]

20 Sep 2024

PONE-D-24-27767

Understanding the pathways to text generation: A longitudinal study on executive functions, oral language, and transcription skills from kindergarten to first grade

PLOS ONE

Dear Dr. Jimenez,

Thank you for submitting your manuscript to PLOS ONE. We have received comments from 2 reviewers. Both of them are positive about the manuscript. But they also made some recommendations to revise the manuscript. Reviewer 2 particularly highlights that the loadings of cognitive flexibility were low and recommends some discussion on this. Reviewer 2 also requests for clarifications on some details.

We look forward to receiving your revised manuscript.

Kind regards,

Yiu-Kei Tsang

Academic Editor

PLOS ONE

Journal Requirements:

2. Thank you for stating the following financial disclosure: “Grant PID2019-108419RB-100 funded by MCIN/AEI/10.130.39/50110001103”

3. Please note that your Data Availability Statement is currently missing the repository name and/or the DOI/accession number of each dataset OR a direct link to access each database. If your manuscript is accepted for publication, you will be asked to provide these details on a very short timeline. We therefore suggest that you provide this information now, though we will not hold up the peer review process if you are unable.

Additional Editor Comments:

The comments by Reviewer 2 are listed below:

Thanks for inviting me to review this paper “Understanding the pathways to text generation: A longitudinal study on executive functions, oral language, and transcription skills from kindergarten to first grade”. Overall, it presented an interesting exploration for extending the Not-so-simple view of writing. Although the author has provided some interesting insight for explaining their finding, I still found that there are much to address and would not be able to recommend this study to be published in the journal “PLoS ONE” at this moment.

1. “Introduction”. Please elaborate on why writing production is a multidimensional phenomenon and how it is connected with the focus of dimensions other than writing quality.

2. “Introduction”. Please explain the necessity of extending Kim et al.’s study. How does research of this type contribute to the field? In the other sections of the manuscript, the authors claim to extend the Not so simple view of writing.

3. “Literature review”. The author has included a substantial amount of evidence in the literature review, giving it the potential to become a solid and informative review. However, I am not very convinced on some arguments such as “ few studies have explored the significance of executive functions” given the substantial studies (even including meta-analysis studies) have done. Meanwhile, since I am not an expert in Spanish, may I know if there is any language that is with shallow orthography, where relevant studies on similar topics have been done?

4. “Methodology”. Please include more detail related to the measurement of syntactic complexity. Instead of common calculations such as TTR, the study assigns weight to different types of sentences. What is the specific rationale for assigning such a weight to the sentence? In addition, please provide an example of each class of sentence and explain why the sentence was identified in a particular way.

5. “Methodology” The authors provided reliability for some tasks. Please provide reliability values for remaining tasks.

6. “Results”. The authors claimed that “all absolute values of skewness and kurtosis remained below 3.0 and 10.0”, however, it is not the truth in Table 1.

7. “Results”. Please include a measurement model before presenting the three models.

8.” Results”. There are some notable issues in the models, for instance, the surprising low loadings for cognitive flexibility across EF latent variables. In addition, I am rather confused in reading the Tables and Figures. For instance, the WTU looks like insignificant in the SC model (dashed line with two digits of decimals, .64), however, the table indicated it was significant (.641, three digits of decimals). I assume the information provided in tables and figures are the same, why is it necessary to present in two different ways?

9. “Discussion and implications”. This study only examined the relationship of writing components (Time point1) and writing performance (Time point2). I doubt that such a design could be claimed as “longitudinal”.

10. “Discussion and implications”. More studies could be linked in the discussion section. For instance, you can compare your findings with previous studies in discussion between lines 771 to 783. Also, the explanation between lines 830 to 835 would be more convincing if the author could provide some reference. This comment is also applicable to the explanation between lines 820 to 846.

11. “Limitations”. This study did not measure students’ non-verbal intelligence (which usually did in EF related studies) nor other linguistic skills (orthographic knowledge/morphological awareness which may relate to the languages you claimed “shallow orthography”). Such limitation should be acknowledged and justified.

We invite you to submit a revised version of the manuscript that addresses the points raised during the review process.

Reviewers' comments:

Reviewer's Responses to Questions

**Comments to the Author**

1. Is the manuscript technically sound, and do the data support the conclusions?

Reviewer #1: Yes

2. Has the statistical analysis been performed appropriately and rigorously? 

Reviewer #1: Yes

3. Have the authors made all data underlying the findings in their manuscript fully available?

Reviewer #1: No

4. Is the manuscript presented in an intelligible fashion and written in standard English?

Reviewer #1: Yes

5. Review Comments to the Author

Reviewer #1: In general, it is a good manuscript, with important practical implications. As a consideration, I would like to know more information abouta participants. I consider that some sociodemographic characteristisc should be considered. In addition, it could be interesting to expand the study to other regions of Spain, in order to obtain more robust conclusions.

6. PLOS authors have the option to publish the peer review history of their article (what does this mean?). If published, this will include your full peer review and any attached files.

Reviewer #1: No

---

## [Author Response · Author response to Decision Letter 0]

21 Nov 2024

We have uploaded the Response to Reviewers document to the journal's online submission system."

---

## [Decision Letter · Decision Letter 1]

2 Dec 2024

Understanding the pathways to text generation: A longitudinal study on executive functions, oral language, and transcription skills from kindergarten to first grade

PONE-D-24-27767R1

Dear Dr. Jimenez,

We’re pleased to inform you that your manuscript has been judged scientifically suitable for publication and will be formally accepted for publication once it meets all outstanding technical requirements.

Kind regards,

Yiu-Kei Tsang

Academic Editor

PLOS ONE

Additional Editor Comments (optional):

Reviewers' comments:

Reviewer's Responses to Questions

**Comments to the Author**

1. If the authors have adequately addressed your comments raised in a previous round of review and you feel that this manuscript is now acceptable for publication, you may indicate that here to bypass the “Comments to the Author” section, enter your conflict of interest statement in the “Confidential to Editor” section, and submit your "Accept" recommendation.

Reviewer #2: All comments have been addressed

2. Is the manuscript technically sound, and do the data support the conclusions?

Reviewer #2: Yes

3. Has the statistical analysis been performed appropriately and rigorously? 

Reviewer #2: Yes

4. Have the authors made all data underlying the findings in their manuscript fully available?

Reviewer #2: Yes

5. Is the manuscript presented in an intelligible fashion and written in standard English?

Reviewer #2: Yes

6. Review Comments to the Author

Reviewer #2: Thank you to the author(s) for their significant efforts in addressing my concerns. I am pleased to recommend this interesting manuscript.

7. PLOS authors have the option to publish the peer review history of their article (what does this mean?). If published, this will include your full peer review and any attached files.

Reviewer #2: No

---

## [Editor Report · Acceptance letter]

9 Dec 2024

PONE-D-24-27767R1 

PLOS ONE

Dear Dr. Jimenez, 

I'm pleased to inform you that your manuscript has been deemed suitable for publication in PLOS ONE. Congratulations! Your manuscript is now being handed over to our production team.

Kind regards, 

on behalf of

Dr. Yiu-Kei Tsang 

Academic Editor

PLOS ONE